# Deletion of Spinophilin Promotes White Adipocyte Browning

**DOI:** 10.3390/ph16010091

**Published:** 2023-01-08

**Authors:** Wenyu Gou, Hua Wei, Lindsay Swaby, Erica Green, Hongjun Wang

**Affiliations:** 1Department of Surgery, Medical University of South Carolina, Charleston, SC 29425, USA; 2Ralph H. Johnson Veterans Affairs Medical Center, Charleston, SC 29401, USA

**Keywords:** spinophilin, white adipocyte browning, PPAR-γ, obesity, UCP-1

## Abstract

Browning of white adipose tissue (WAT) is suggested as a promising therapeutic approach to induce energy expenditure and counteract obesity and its associated complications. Systemic depletion of spinophilin (SPL) increases metabolism and improves energy balance in mice. In this study, we explored the mechanistic insight of SPL action in WAT browning. Gene expression and mitochondria tracker staining showed that visceral white adipose tissue (vWAT) harvested from SPL KO mice had a higher expression of classic browning-related genes, including uncoupling protein 1 (*UCP1*), Cell death inducing DFFA like effector A *(CIDEA)* and PR domain containing 16 *(PRDM16)*, as well as a higher mtDNA level compared to vWAT from wild type (WT) control mice. When adipogenesis was induced in pre-adipocytes harvested from KO and WT mice ex vivo using the PPAR-γ agonist rosiglitazone (Rosi), SPL KO cells showed increased browning marker gene expression and mitochondria function compared to cells from WT mice. Increased PPAR-γ protein expression and nucleus retention in vWAT from SPL KO mice after Rosi treatment were also observed. The effect of SPL on vWAT browning was further confirmed in vivo when WT and KO mice were treated with Rosi. As a result, SPL KO mice lost body weight, which was associated with increased expression of browning maker genes in vWAT. In summary, our data demonstrate the critical role of SPL in the regulation of WAT browning.

## 1. Introduction

Obesity has doubled in global prevalence and is a major public health concern [1]. Furthermore, obesity is associated with an increased risk of diabetes, hypertension, cancer, and cardiovascular disease, as well as a reduced quality of life and profound social and economic costs [2]. Unfortunately, despite the urgent need to address the obesity epidemic, there is a lack of effective and low-risk treatment strategies to counter this issue [3].

Adipose tissue, which may account for 80% of body weight in obese individuals, is composed of both white adipose tissue (WAT) and brown adipose tissue (BAT) [4]. WAT stores excess energy and functions as an endocrine organ, while BAT performs adaptive thermogenesis to maintain homeostasis [5]. BAT is characterized by both increased numbers of mitochondria and high expression levels of UCP1 [6]. UCP1 uncouples mitochondrial respiration from adenosine-5′-triphosphate (ATP) production, erasing a negative feedback loop, and thus allowing for high rates of fatty acid oxidation that produce heat [7]. Because of its ability to dissipate energy through thermogenesis, BAT has strong anti-obesity and anti-diabetic effects [8]. High BAT activity is correlated with a low body mass index and its activation improves aspects of cardiovascular health [9]. Studies have shown that white adipocytes can be converted into “beige cells” through a process known as “browning”, which occurs in response to cold or exercise [4,10,11]. Beige cells demonstrate similar characteristics to BAT, such as a high UCP1 expression [12]. In animal models, induction of browning can successfully treat obesity and insulin resistance, and increased BAT mass appears to reset metabolism and provide long-term control of body weight [13,14]. While the ability to induce browning is of great interest as a potential therapy for obesity and associated diseases [3], the mechanism of browning is not fully understood.

Spinophilin (SPL) is a multidomain scaffolding protein that can modulate the activity of various G-protein-coupled receptors in the central nervous system [15]. SPL interacts with over 30 partner proteins to regulate neuronal signaling, G-protein signaling, membrane receptors, and ion channels [16,17]. Its binding partners include protein phosphatase-1 (PP1), a ubiquitous serine/threonine phosphatase that modulates cell cycle progression, protein synthesis, and transcription [16]. SPL additionally acts as a negative regulator of β-cell M3 muscarinic receptors, which maintain glucose homeostasis throughout the body [15]. SPL knockout mice showed reduced body weight associated with decreases in percentages of body fat mass compared with wild-type mice [15]. A previous study from our lab showed the importance of SPL in WAT browning as ablation of SPL in mice promoted browning and provided protection from high-fat diet-induced obesity and insulin resistance [18].

Peroxisome proliferator-activated receptor-gamma (PPAR-γ), a ligand-activated hormone receptor, is necessary for adipocyte differentiation and is known to promote browning [19,20]. PPAR-γ is expressed selectively in adipose tissues of humans and rodents [20], and it is a key regulator of fat storage and energy homeostasis [21]. PPAR-γ functions by translocating to the nucleus, where it heterodimerizes with retinoid X receptors (RXR) and alters the expression of specific genes. Increased PPAR-γ expression upregulates the expression of BAT marker genes and protects against insulin resistance [19].

Here, we demonstrate that SPL ablation amplifies rosiglitazone, a PPAR-γ agonist, and induced brown-like adipocyte generation in the visceral white adipose tissue (vWAT) depots as identified by UCP1 expression both in vitro and in vivo. We further show that the deletion of SPL led to increased protein expression and nuclear presence of PPAR-γ. SPL negatively controls rosiglitazone-induced browning by blocking the nuclear translocation of PPAR-γ and the consequent binding to the UCP1 promoter.

## 2. Results

### 2.1. Systemic Deletion of SPL Results in Dramatically Lower Body Weights and Lower White Adipose Tissue Mass in Mice

First, we measured the body weights of standard chow-fed SPL KO mice and their wild type (WT) littermate controls from 3 until 20 weeks of age. SPL KO mice exhibited lower body weights compared to age-matched WT controls starting from 8 weeks of age (n = 8–10, Figure 1A). At the age of 10 weeks, WT mice had an average body weight of 25.2 ± 0.5 g, while SPL KO mice had an average body weight of 22.6 ± 0.7 g (n = 19–21, *p* = 0.01 vs. control, Figure 1B,C). The body weight difference was more dramatic at the age of 20 weeks (Figure 1A). Next, we compared the BAT, visceral WAT (vWAT), and inguinal WAT (iWAT) weights from KO and WT. The BAT mass and BAT weight/body weights were similar between SPL KO and WT mice (Figure 1D,E). In contrast, SPL KO mice had significantly smaller vWAT and iWAT weights and fat/body weights compared to corresponding WAT from WT controls (Figure 1D,E). These results demonstrate that SPL deletion reduces obesity and white adipose tissue masses.

### 2.2. Increased Browning Marker Genes Expression and Mitochondria Content in vWAT from SPL KO Mice Compared to WT Controls

We suspect that WAT browning contributes to reduced white adipose mass. Due to the large reduction in vWAT weight, and our previous study which showed that the expression levels of browning markers were not significantly different in iWAT between SPL KO and WT mice [18], we focused on investigating the browning marker gene expression, mitochondria content number, and mitochondrial activity in vWAT from SPL KO and WT mice. We measured the expression of several browning marker genes, including *CIDEA, PGC-1a, UCP1, PPAR-γ, FABP4, and PRDM16*, in BAT and WAT from SPL KO and WT mice at 10 weeks of age. Expression of *UCP1,* the typical marker of brown fat, showed the highest expression level in brown adipose tissue, although there was no dramatic difference between BAT from SPL KO or WT mice (Figure 2A). vWAT from SPL KO mice showed significantly increased mRNA expression levels of *UCP1, FABP4*, and *PRDM16* (Figure 2B), compared to those from WT mice. In addition, compared to vWAT from WT mice, vWAT from SPL KO mice contain significantly more mitochondria, as indicated by increased mtDNA/nDNA levels (Figure 2C), a stronger fluorescent signal of MitoTracker (Figure 2D), and a higher expression of the *CIDEA* gene (Figure 2E). This suggests increased mitochondrial activity in vWAT from SPL KO mice compared to those from WT mice.

### 2.3. Ablation of SPL Potentiates the Differentiation of vWAT SVF toward Brown-like Adipocytes

We isolated primary SVF cells from vWAT from SPL KO or WT mice and induced them into brown adipocytes in vitro by treatment with the browning induction media and then measured the expression of browning maker genes. SVF-derived adipocytes from SPL KO mice showed significantly higher mRNA levels of the browning genes including *CIDEA, UCP1.* The adipocyte differentiation marker and PPAR-γ responsive gene, *FABP4* was increased as well. There was no significant difference in the expression of *PPAR-γ* and *PRDM16* (Figure 3A). SPL KO SVF-adipocytes stained with Oil Red appeared to have more dense and smaller lipid droplets which are characteristics seen in brown adipocyte staining (Figure 3B). Consistently, UCP1 protein was significantly increased in fully differentiated SVF-adipocytes (Figure 3C). Furthermore, the basal and uncoupled OCR and the mitochondrial activity were significantly increased in the differentiated SVF of SPL KO mice compared to those from the controls (Figure 3D,E), suggesting that vWAT from SPL KO mice were more prone to being induced into brown-like adipocytes.

To confirm these data, we generated SPL knockdown cells by transfecting 3T3L1 cells with the shRNA for SPL. We then induced browning in 3T3L1 or control cells and measured gene expression of *UCP1, FABP4*, and *PPAR-γ*. Like SVF-derived adipocytes, there were significantly more browning maker gene expressions in SPL knockdown 3T3L1 cells (Figure 3F). These results support that the expression of SPL negatively regulates the browning of vWAT.

### 2.4. SPL Ablation Favors vWAT Browning through Promoting Nuclear Translocation/Presence of PPAR-γ

To explore the potential mechanism of SPL ablation in vWAT browning, we measured the expression and cellular distribution of PPAR-γ, a ligand-dependent transcription factor that regulates UCP1 expression in SVF-derived pre-adipocytes from SPL KO or WT mice treated with rosiglitazone (2 µM). Cells were cultured for 2 days with an induction medium followed by high doses of rosiglitazone at 2 μM for another 48 h for further browning induction. We measured the cytosolic and nuclear presence of PPAR-γ at 1 h, 2 h, and 48 h after incubation with 2 μM rosiglitazone. In both the WT and KO cells, increased *PPAR-γ* was present in both the nucleus and cytosol. PPAR-γ expression levels were slightly increased in both nucleus and cytosol for WT cells at 1 h and 2 h after treatment and decreased at 48 hours after treatment (Figure 4A). In contrast, there was a much higher PPAR-γ level in both the nucleus and cytoplasm fraction in SPL KO cells compared to WT cells. Furthermore, higher levels of PPAR-γ were detected in the nuclear portion of the SPL KO cells at all the time points measured (Figure 4A). PPAR-γ nucleus retention was further confirmed by immunofluorescence staining, in which more PPAR-γ was present in the nucleus of KO cells (Figure 4B). Consequently, the blockage of PPAR-γ activity using GW9662 led to the reduced expression of *UCP1*, but not *FABP4* (Figure 5A,B), suggesting that PPAR-γ activity is required for UCP1 activation. In conclusion, this data suggests that SPL is a potential regulator for PPAR-γ nuclear translocation, and SPL ablation retains PPAR-γ in the nucleus and further promotes the white-to-brown fat conversion.

### 2.5. Ablation of SPL Promotes vWAT Browning Induced by PPAR-γ Agonist In Vivo

To further assess the impact of SPL on WAT browning in vivo, WT and SPL KO mice were treated with rosiglitazone (10 mg/kg daily for 10 days, i.p.). Mice from both groups showed reduced body weight gain after receiving rosiglitazone. In contrast to WT, SPL KO mice had body weight loss compared to before treatment (Figure 6A). Like the in vitro condition, treatment with rosiglitazone led to an increase in browning as indicated by enhanced expression of UCP1 in the vWAT of both WT and SPL KO mice. Furthermore, there were significantly more expressions of *CIDEA, PRDM16,* and *UCP1* in vWAT from SPL KO mice compared to vWAT from WT mice after rosiglitazone treatment (Figure 6B). Increased *UCP1* gene expression was further demonstrated by immunofluorescent staining of UCP1 in vWAT tissue section (Figure 6C). These data indicate that robust browning occurred in vWAT from SPL KO mice after rosiglitazone treatment, which might have contributed to reduced body weights.

## 3. Discussion

We have shown in our previous study that SPL knockout mice are protected from diet-induced obesity and insulin resistance, at least in part, by promoting the browning of white adipocytes [18]. This was associated with increased browning in vWAT from SPL KO mice compared to WT littermate controls. Our current study further deciphered the role of SPL in vWAT browning and explored the molecular mechanisms of how ablation of SPL leads to increased browning.

Our results show that the deletion of SPL leads to the induction of expressions of brown adipocyte-related genes, including *UCP1, FABP4,* and *CIDEA*, specifically in vWAT collected from SPL KO mice. In vitro, we showed that SVF-derived adipocytes from SPL KO mice or SPL knockdown 3T3L1 cells were prone to browning induction when stimulated with rosiglitazone. Further mechanistic studies demonstrated that higher *UCP1* expression in SPL KO cells was associated with the induction and nuclear tentation of PPAR-γ after rosiglitazone treatment. These data suggest that SPL negatively controls *UCP1* expression via *PPAR-γ, CIDEA*, *PRDM16*.

One mechanism of increased *UCP1* in SPL KO adipocytes is through potentiating the effects of PPAR-γ. As a transcription factor, nucleus translocation and retaining of PPAR-γ is critical for its function in promoting *UCP1* gene transcription [22]. SPL knock-out vWAT stimulated with rosiglitazone ex vivo showed higher expressions of several key feature molecules involved in PPAR-γ-mediated WAT browning, including *CIDEA*, *PGC-1a,* and *PRMD16,* compared to WAT from control mice. Studies have shown that PPAR-γ dynamically shuttles between the nucleus and cytoplasm, although they constitutively and predominantly appear in the nucleus [23]. Treatment with PPAR-γ agonist improve human adipocyte metabolism [24]. For example, rosiglitazone has been proven to remodel the lipid droplet and “brown” human visceral and subcutaneous adipocytes ex vivo by coordinating structural and metabolic remodeling in adipocytes [25]. When rosiglitazone, a PPAR-γ agonist, was used to induce PPAR-γ activation in white adipocyte browning in this study, not only sustained PPAR-γ expression was observed, but also it remained in the nucleus for a longer time, which indicates that SPL controls the nuclear-cytoplasmic transport regulation of PPAR-γ. When SPL is deleted, PPAR-γ expression is increased and it remains in the nucleus, constantly inducing *UCP1* gene expression.

Based on our data, the loss of SPL leads to an increased expression of PPAR-γ in both the cytosol and nuclei of vWAT adipocytes. This was the case in the quiescent stage as well as after treatment with Rosi, suggesting that SPL itself negatively regulates the expression level of PPAR-γ. The degradation of PPAR-γ after ligand activation is achieved via the ubiquitination-proteasome pathways [26]. Our data also showed that there was much more PPAR-γ remaining inside the nucleus of SPL KO cells at 48 hours after Rosi treatment compared to WT cells, indicating that SPL is required for PPAR-γ degradation. Indeed, SPL was shown to directly regulate protein stability and/or ubiquitination in a cyclin-dependent kinase 5 (Cdk5)-dependent manner in Parkinson’s disease [27]. The ERK/Cdk5 axis also controls PPAR-γ function in type 2 diabetes [28]. Therefore, the loss of SPL preserved PPAR-γ protein stability and blocked its ubiquitination/degradation in the KO cells, likely in a Cdk5-dependent manner. However, further studies are needed to definitively reveal this mechanism.

We also show that PPAR-γ activity is required for *UCP1* induction. Blocking PPAR-γ activity by GW9662 partially reduced *UCP1* expression, suggesting that PPAR-γ independent effects also mediate browning of beige adipocytes from white adipose pre-adipocytes and even from most developmentally epididymal WAT depot. This has been confirmed in vivo as SPL KO mice treated with rosiglitazone showed more browning gene expression and body weight loss.

CIDEA is also a potential downstream target of SPL action on browning. For example, KO of SPL likely leads to upregulated *UCP1* expression through the upregulation of *CIDEA*. CIDEA, a lipid-droplet-associated protein enriched in brown adipocytes promoting the enlargement of lipid droplets, has been shown to maintain the capability of human adipocytes britening/beiging in thermogenesis [25,29], and knockout of *CIDEA* suppresses *UCP1* expression [30]. During browning/beiging induction, CIDEA shuttles from lipid droplets to the nucleus and specifically inhibits LXRα repression of *UCP1* enhancer activity and strengthens PPAR-γ binding to *UCP1* enhancer, hence driving *UCP1* transcription. Therefore, SPL ablation likely enhanced *UCP1* expression via the upregulation of CIDEA.

Another potential mechanism of SPL action in vWAT is through the regulation of PRDM16. PPAR-γ ligands require the expression of *PRDM16*, a factor that controls the development of classical brown fat for white adipose browning [31,32]. PRDM16 and rosiglitazone synergistically activate the brown fat gene program in vivo [33]. PRDM16 stimulates brown adipogenesis by binding to PPAR-γ and activating its transcriptional function [34]. An increase in PRDM16 expression in SPL KO cells might lead to an increase in the transcriptional function of PPAR-γ, and consequently, increased UCP1 expression.

SPL systemic KO mice were used for the study. To avoid the systemic effects, we isolated vWAT and tested the effect ex vivo. Therefore, the effects we observed in vWAT seemed to be tissue-specific effects. However, the obesity reduction effects induced by rosiglitazone in SPL KO mice were likely not only caused by vWAT browning, since mice had reduced whole body weight after treatment. Further mechanistic studies using tissue-specific knockout mice are needed to define the role of SPL in controlling WAT browning and may help explain the phenomenon observed. Additionally, whether SPL plays a role in thermogenesis is also an interesting topic for future studies.

One limitation of this study is that SPL is highly enriched in dendritic spines, whether it functions via neuroendocrine signaling and has potential secondary effects on browning has not been studied in this paper. In summary, our data support that the depletion of SPL in vWAT leads to increased white adipocyte browning in vitro and in vivo. Targeting SPL may be used to promote WAT browning and reduce obesity. Another limitation is that we did not test whether overexpression of SPL would impact rosiglitazone-induced browning.

## 4. Materials and Methods

### 4.1. Animal

Whole-body SPL KO mice were originally obtained from Dr. Ning Wang’s lab at the University of Alabama at Birmingham and mouse genotyping was performed as described previously [18]. Mice were maintained under a standard 12:12 h light-dark cycle. Food and water were available ad libitum. Only male mice were used in the study since females did not show differences in body weight when fed a high-fat diet [18]. All animal experiments were approved by the Animal Care Committee at the Medical University of South Carolina (IACUC, protocol #: 200972).

#### Tissue Collection and Stromal Vascular Fraction (SVF) Pre-Adipocyte Isolation

BAT, visceral WAT (vWAT), and inguinal WAT (iWAT) were dissected and weighted. To collect SVF-derived pre-adipocytes, adipose tissue was digested with a mixture of 1.5 U/mL of collagenase D (Roche), 2.4 U/mL of Dispase II (Roche), and 10 mM of CaCl_2_ at 37 °C for 40–50 min. Cells were washed, seeded on a collagen-coated dish, and then cultured in a medium including DMEM/F12 supplemented with 10% fetal bovine serum (FBS), 1% penicillin, and streptomycin (complete medium) in 5% CO_2_ at 37 °C before further differentiation [35].

### 4.2. Adipocyte Differentiation and Browning Induction

Pre-adipocytes from SVF or 3T3L1 cells (ATCC) were cultured in the complete medium until 80% confluence. Cells were induced to differentiate into beige/brite cells using a three-step protocol as reported [35,36]. First, cells were cultured in an induction medium that contained DMEM/F12 base media and 125 µM indomethacin (Sigma-Aldrich, St. Louis, MO, USA-Aldrich, St. Louis, MO, USA), 2 µg/mL dexamethasone (Sigma-Aldrich, St. Louis, MO, USA), 0.5 mM IBMX (Sigma-Aldrich, St. Louis, MO, USA), 5 µg/mL Insulin, 1 nM triiodothyronine (T3; Sigma-Aldrich, St. Louis, MO, USA), and 0.5 µM rosiglitazone (Sigma-Aldrich, St. Louis, MO, USA) for 48 h (days 1–2). Second, cells were switched to a maintenance medium I that contained 5 µg/mL of insulin, 1 nM thyroid hormone (T3), and 0.5 µM rosiglitazone for another 48 h (days 3–4). Third, cells were cultured into the maintenance medium II, which contained 5 µg/mL Insulin, 1 nM T3, and 1 µM rosiglitazone for 48–72 h (days 5–7) [35]. After full differentiation, cells were collected for RNA extraction and gene expression analysis. Oil Red O (Sigma-Aldrich, St. Louis, MO, USA-O0625) staining was used to identify oil drops in cells according to the manufacturer’s instructions.

### 4.3. Real-Time PCR Analysis

Total RNA was extracted using a Qiagen, Hilden, Germany RNA kit (Qiagen, Hilden, Germany). Samples were treated with DNase (Sigma-Aldrich, St. Louis, MO, USA) to exclude DNA contamination. The concentration and quality of RNA were determined according to optical density (OD) measurement. mRNA was reverse transcribed into cDNA using an RT-PCR kit (Bio-Rad, Hercules, CA, USA). Advanced Universal SYBR Green Supermix was used for quantitative RT-PCR in a CFX96 real-time thermocycler (Bio-Rad, Hercules, CA, USA) to determine gene expression levels of *UCP1*, *PGC-1a, PRDM16, FABP4, CIDEA, PPAR-γ*. The primer sequences are listed in Table 1. Fold changes in gene expression was normalized to *β-actin* expression and were plotted and compared between groups.

### 4.4. Mitochondrial DNA Content

Total DNA was extracted using the DNeasy mini kit (Qiagen, Hilden, Germany) and was then quantified using the Pico green dsDNA quantification kit (Invitrogen, Waltham, MA, USA). The mitochondrial genome (mtDNA) and nuclear DNA (nDNA) Ct levels were obtained from RT-PCR and the relative mtDNA content was calculated as 2 × 2 ^Δ(nDNA Ct − mtDNA Ct)^ [37,38].

### 4.5. Mitochondria Membrane Potentials (MMP)

Mitochondrial membrane potentials were analyzed using the tetramethylrhodamine ethyl ester (TMRE) Mitochondrial Membrane Potential Assay Kit (Cell Signaling Technology, Danvers, MA, USA). In brief, 200 nM of TMRE was added to the medium of the differentiated SVF-derived adipocytes and incubated at 37 °C for 20 min. After washing with PBS twice, the confocal images of cells were captured by the Leica TCS SP5 confocal laser scanning microscope.

### 4.6. Oxygen Consumption Rates (OCR)

OCR for fully differentiated SVF-derived adipocytes was assessed with a Seahorse XFe96 analyzer (Agilent Technologies, Santa Clara, CA, USA). Cells were washed and maintained with the XF assay media in a non-CO_2_ incubator at 37 °C for 1 h. During the incubation time, 25 µL of 10 µM oligomycin, 9 µM carbonyl cyanide-p-trifluoromethoxyphenyl-hydrazone (FCCP), 10 µM rotenone, and 10 µM antimycin A were added to the injection ports of the XFe96 sensor cartridge. Seventy minutes after starting the experiment, the instrument injected the inhibitors into the cells. OCR was measured continuously.

### 4.7. Deletion of SPL Gene in 3T3L1 Cells by shRNA Transfection

3T3L1 cells were cultured in a complete medium until 80% confluence. shRNA for SPL or a control shRNA was mixed with plasmid transfection reagent and incubated at room temperature for 45 min according to the manufacturer’s instruction (Santa Cruz Biotechnology, Dallas, TX, USA). The mixture was added dropwise into 3T3L1 cells and incubated in the 37 °C CO_2_ incubator for 8 h. Complete medium was added and cultured for another 24 h. Transfected cells were then selected using media containing puromycin for 3 days. Surviving cells were used for further experiments.

### 4.8. Western Blot Analysis

Cytosolic and nuclear fractions of cells were separated using the Cell Fraction Kit (Abcam, Cambridge, UK). Nuclear or cytosolic proteins were separated by SDS-PAGE, transferred to polyvinylidene difluoride membranes, and incubated with primary antibodies against UCP1 (Sigma-Aldrich, St. Louis, MO, USA, cat# U6382), PPAR-γ (Thermo Fisher, Waltham, MA, USA, cat# ma5-14889), GAPDH (Sigma-Aldrich, St. Louis, MO, USA, cat# G8795), Histone H3 (Cell Signaling Technology, Danvers, MA, USA, cat# 4499P) or Lamin A (Abcam, Cambridge, UK, cat# ab8980). The membrane was then washed and incubated with the corresponding horseradish peroxidase-conjugated secondary antibodies (Cell Signaling Technology, Danvers, MA, USA Technology). Signals were visualized using an ECL detection kit (Thermo Scientific, cat# 34096). The relative expression of proteins was quantified using the ImageJ software.

### 4.9. Immunofluorescence Staining

Mouse adipose tissues were fixed in 10% neutral buffered formalin overnight, embedded in paraffin, and then sectioned with a thickness of 5 μm each. Sections were stained with the rabbit anti-UCP1 antibody (Sigma-Aldrich, St. Louis, MO, USA, cat# U6382, 1:200 in dilution). SVF-derived adipocytes or 3T3-L1 cells were cultured on a coverslip and fixed in cold acetone for 10 min. Sections were stained with a rabbit anti-PPAR-γ antibody (Thermo Fisher, Waltham, MA, USA, cat# ma5-14889, 1:200 in dilution) overnight at 4 °C. Slides were washed with PBS, stained with Alexa Flour 488 goat anti-rabbit (Life Tech, cat#A11008, 1:500 dilution) secondary antibody, washed, covered with a coverslip, and observed under a fluorescence microscope.

### 4.10. Statistical Analysis

Data were expressed as mean ± standard error of the mean (SEM). Differences between groups were compared for statistical significance by one-way or two-way ANOVA with Tukey-Kramer post hoc test or *Student’s t*-test using GraphPad Prism 8.0 (GraphPad Inc, CA, USA). Other specific tests used are referenced in figure legends with *p* < 0.05 denoting significance.

## Figures and Tables

**Figure 1 pharmaceuticals-16-00091-f001:**
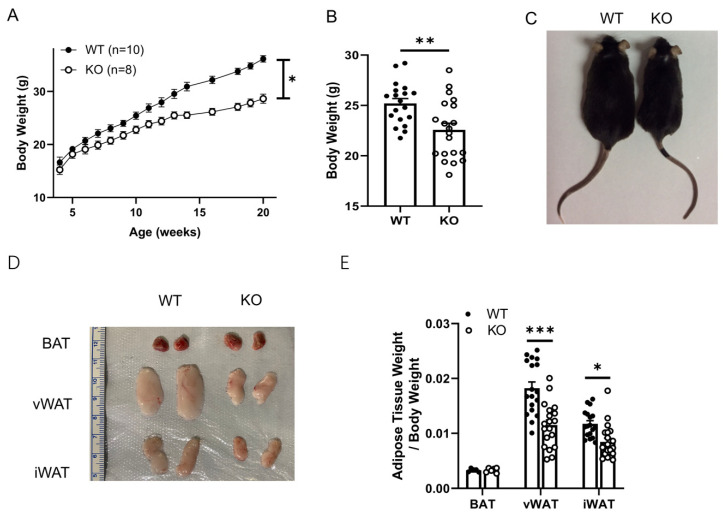
**SPL KO mice showed smaller body weights and WAT mass compared to littermate controls.** (**A**). Average body weights of SPL KO (KO, n = 8) and wild-type littermate controls (WT, n = 10) at different ages. (**B**). Body weights and (**C**) representative photo of SPL KO and WT mice at 10 weeks of age. (**D**). Represent micrograph of BAT, vWAT, and iWAT of SPL KO and WT at 10 weeks of age (**E**). Adipose tissue weight/body weights of SPL KO and WT mice. B & E: n = 19–21 per group. Unpaired *Student’s t*-test was performed to compare the difference between WT and KO. * *p* < 0.05, ** *p* < 0.01, and *** *p* < 0.001. All data are presented as means ± SEM.

**Figure 2 pharmaceuticals-16-00091-f002:**
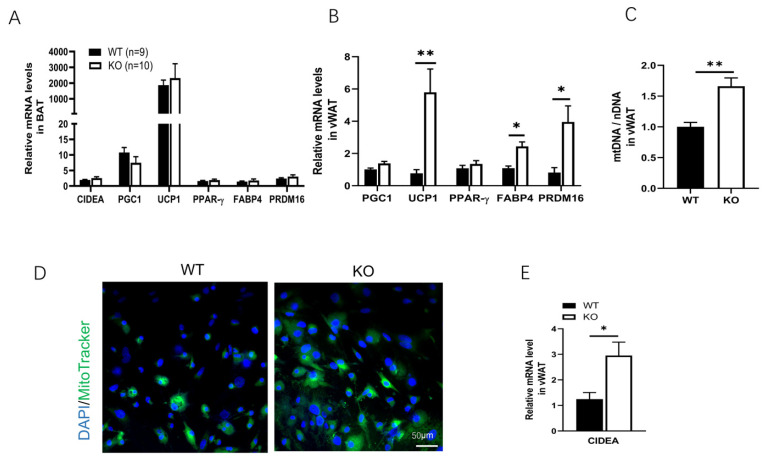
**WAT from SPL KO mice showed higher browning-related gene expression and increased mt DNA level.** (**A**) Expression of *CIDEA, PGC-1a, UCP1, PPAR-γ, FABP4,* and *PRFM16* relative to *beta-actin* in brown adipose tissue (BAT) and vWAT (**B**). (**C**) Relative mitochondria DNA (mtDNA/nDNA) levels in vWAT from WT and KO mice. (**D**) Mitochondria tracker staining of mitochondria in vWAT from WT and KO mice. Green: mitochondria, blue: nuclei. (**E**) Relative expression of *CIDEA* mRNA in vWAT from WT and KO mice. 10 weeks old animal experiments were performed with n = 9–10 mice in each group. Unpaired *Student’s t*-test was performed to compare the difference between WT and KO. * *p* < 0.05, ** *p* < 0.01. All data are presented as means ± SEM.

**Figure 3 pharmaceuticals-16-00091-f003:**
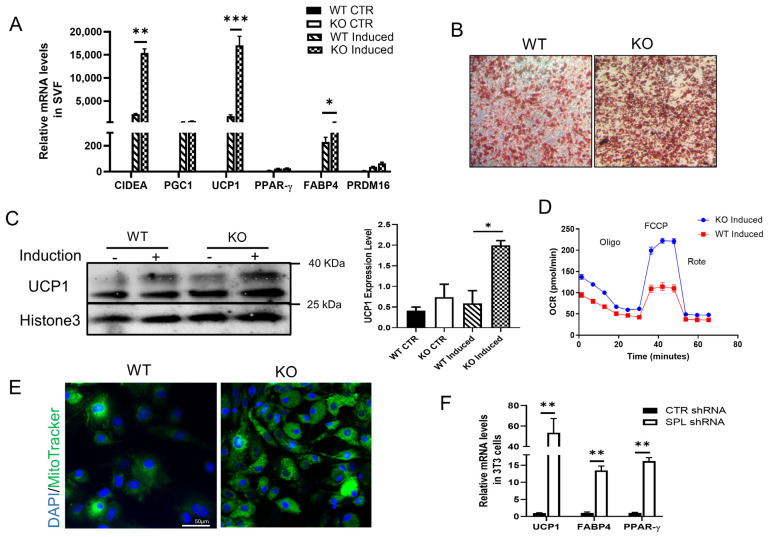
**In vitro browning induction showed increased browning gene expression in SVF-derived adipocytes from WT and KO mice.** vWAT were harvested from WT and KO mice, cultured, and treated with a full browning induction medium to induce browning, and expression of markers was measured. (**A**) mRNA expression of *CIDEA, PGC1, UCP1, PPAR-γ,* and *FABP4* relative to beta-actin in SPL KO and WT adipocytes. (**B**) Oil-red staining of SVF after full browning induced. (**C**) UCP1 protein expression level in vWAT cells from WT or KO mice after full browning induced. Relative protein expression of UCP1 compared to the expression of Histone 3 in vWAT induced. (**D**) Oxygen consumption rate (OCR) and (**E**) mitochondria tracker in WT and KO mitochondria function after browning induction. (**F**) Relative mRNA levels of *UCP1, FABP4,* and *PPAR-γ* in control or SPL shRNA transfected 3T3L1 cells after browning induction. Data were from at least three individual experiments. Two-way ANOVA with Tukey-Kramer post hoc test was used to compare the difference in (**A**,**C**). Unpaired *Student’s t*-test was performed to compare the difference between WT and KO in (F). * *p* < 0.05, ** *p* < 0.01, and *** *p* < 0.001. All data are presented as means ± SEM.

**Figure 4 pharmaceuticals-16-00091-f004:**
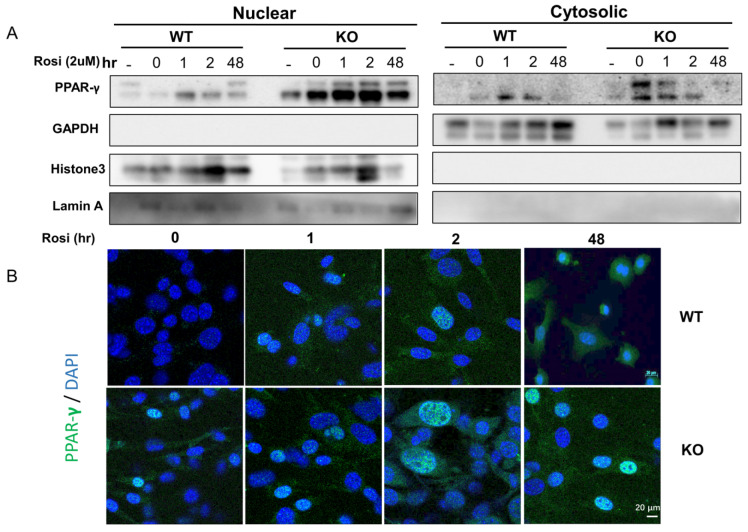
**SPL KO vWAT adipocytes promoted/retained more PPAR-γ in the nuclear after Rosiglitazone stimulation.** Browning was induced in SVF vWAT adipocytes for 2 days and then treated with rosiglitazone at 2μM for another 48 h. (A) Western blot of PPAR-γ, GAPDH, Histone 3, and Lamin A in the nuclear fraction or cytosol fraction of vWAT harvested from WT and SPL KO mice before (0 h) and at 1 h, 2 h, and 48 h post-rosiglitazone treatment. (**B**). Immunofluorescent staining of PPAR-γ in WT and KO mice before (0 h) and 1 h, 2 h, and 48 h post Rosiglitazone treatment. Data were from at least three individual experiments. Scale bar = 20 μm.

**Figure 5 pharmaceuticals-16-00091-f005:**
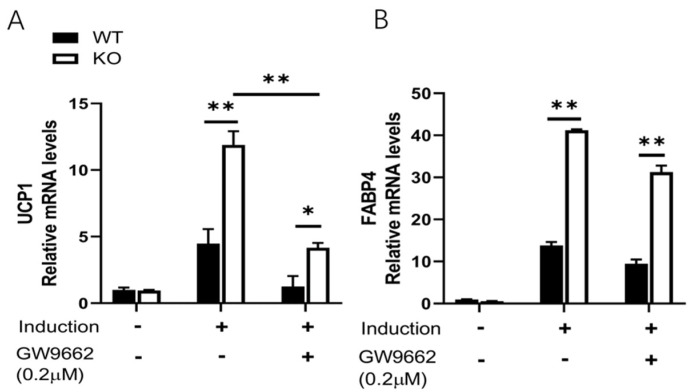
**Blocking PPAR-γ activity reduces rosiglitazone-induced UCP-1 expression in SPL KO adipocytes.** Relative expression of *UCP1* (**A**), and *FABP4* (**B**) in the presence or absence of rosiglitazone (induction) and/or GW9662. SVF cells were pre-treated with 0.2 μM GW9662 for 6 h, then cultured in browning induction media. Two-way ANOVA with Tukey-Kramer post hoc test was performed to compare differences. * *p* < 0.05, ** *p* < 0.01. All data are presented as means ± SEM.

**Figure 6 pharmaceuticals-16-00091-f006:**
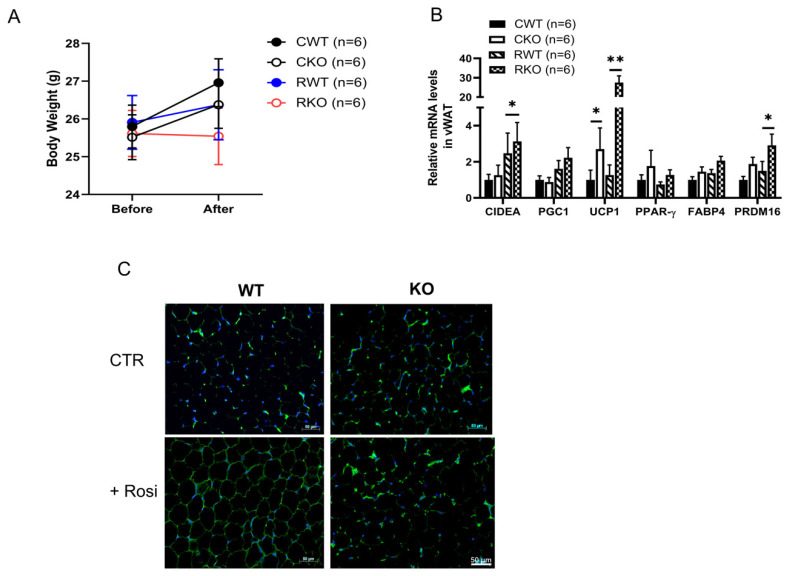
**Treatment with rosiglitazone led to bodyweight reduction and vWAT browning in SPL KO compared to WT mice.** WT and KO mice were treated with rosiglitazone at 10 mg/kg daily for 10 days. (**A**) Body weight changes in WT mice receiving vehicle (CWT), rosiglitazone (RWT), or SPL KO mice receiving vehicle (CKO) or rosiglitazone (RKO). N = 6 mice per group. (**B**) Expression of browning marker genes at mRNA levels in vWAT tissues from SPL KO and WT mice. (**C**) Immunofluorescent staining of UCP1 in vWAT from rosiglitazone treated WT or SPL KO mice. UCP1: green. Scale bar = 50 μm. 2-way ANOVA with Tukey-Kramer post hoc test was performed to compare differences. * *p* < 0.05, ** *p* < 0.01. All data are presented as means ± SEM.

**Table 1 pharmaceuticals-16-00091-t001:** Primer sequences used for RT-PCR analysis.

*UCP1* forward:	5′-CTTTGCCTCACTCAGGATTGG-3′
*UCP1* reverse:	5′-ACTGCCACACCTCCAGTCATT-3′
*PCG-1*a forward:	5′-AGCCGTGACCACTGACAACGAG-3′
*PCG-1*a reverse:	5′-GCTGCATGGTTCTGAGTGCTAAG-3′
*PRDM16* forward:	5′-CCACCAGCGAGGACTTCAC-3′
*PRDM16* reverse:	5′-GGAGGACTCTCGTAGCTCGAA-3′;
*FABP4* forward:	5′-ACACCGAGATTTCCTTCAAACTG-3′
*FABP4* reverse:	5′-CCATCTAGGGTTATGATGCTCTTCA-3′;
*CIDEA* forward:	5’-TGCTCTTCTGTATCGCCCAGT-3’
*CIDEA* reverse:	5’-GCCGTGTTAAGGAATCTGCTG-3’
*PPAR-γ* forward:	5′-CTGTCGGTTTCAGAAGTGCC-3′
*PPAR-γ* reverse:	5′-ATGGTGATTTGTCCGTTGTC-3′

## Data Availability

Original data generated and analyzed during this study are included in this published article or in the data repositories listed in References.

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
