# Peer review of "Deletion of Spinophilin Promotes White Adipocyte Browning"

_pharmaceuticals, 2023, doi:10.3390/ph16010091_

Round 1
Reviewer 1 Report
In the current manuscript, Guo et al. claim that Spinophilin (SPL) is a novel negative regulator of white adipocyte browning via PPAR-γ. The authors conclude that deletion of SPL in mice results in dramatic loss of body weight and white adipose tissue (WAT) (Figure 1). However, the conclusion that SPL acts as a novel negative regulator in energy metabolism, balance caused by white fat browning, may be due to whole body knockout system or fibrosis by cell death. Although the same group reported that SPL-deficient mice are protected from diet-induced obesity and insulin resistance (Zhang et al 2020), it is necessary to characterize SPL whole-body KO mice in terms of other tissues or other phenotypes in current manuscript. These evidences help highlight the role of SPL in WAT function.
1. The authors may provide clearer data to clarify whether SPL affects food intake, brain function, or behavior.
2. The authors may check whether SPL affects fat tissue fibrosis or cell death of white preadipocyte.
3. The authors can determine overall level of SPL expression level for tissue distribution. They need rationale for the role of SPL in white fat.
4. Is SPL specific for WAT but not brown adipose tissue (BAT)? The authors can provide clearer data to clarify whether SPL affects only WAT browning but not interscapular BAT browning.
5. The authors can do cold tolerance test for SPL KO mice if SPL plays a critical role in thermogenesis.
6. I suggest that the authors exclude PPAR-γ from their conclusions since they don’t have molecular mechanism between SPL and PPAR-γ. For example, the manuscript’s title can be “Deletion of Spinophilin Promotes White Adipocyte Browning”.
Author Response
- The authors may provide clearer data to clarify whether SPL affects food intake, brain function, or behavior.
In our published study (Zhang, et al, AJP Endocrinol Metab., 2020, Aug1; 319(2): E354-E362, reference #3 in the current paper), we have compared activities, including behavior, food intake, drinking and licking, body temperature of SPL KO and wild type littermate control mice fed normal chow or high-fat diet (HFD). Analysis of food intake and caloric intake showed that SPL KO mice fed either chow or an HFD ate somewhat more than corresponding controls, suggesting that reduced obesity of male SPL KO mice was not a consequence of reduced food intake. Both male and female SPL KO mice exhibited increased locomotor activity compared to WT controls, however, no differences were observed in HFD-fed SPL KO and WT female mice. No significant differences in body temperature were observed in either males or females.
- The authors may check whether SPL affects fat tissue fibrosis or cell death of white preadipocytes.
This is a thoughtful suggestion, so far we did not observe fat tissue fibrosis or cell death in our study, but this possibility can be further studied in more detail and data will be included in future publications.
- The authors can determine overall level of SPL expression level for tissue distribution. They need rationale for the role of SPL in white fat.
Based on the data from Integrated Proteomics, SPL protein is overexpressed in the brain and peripheral mononuclear cells, adrenal, testis, etc. As stated in the Introduction, the rationale for studying the role of SPL is the following: “SPL knockout mice showed reduced body weight associated with decreases in percentages of body fat mass compared with wild-type mice (15). A previous study from our lab showed the importance of SPL in WAT browning as ablation of SPL in mice promoted browning and provided protection from high fat -diet-induced obesity and insulin resistance (18). Therefore we performed the current study.
- Is SPL specific for WAT but not brown adipose tissue (BAT)? The authors can provide clearer data to clarify whether SPL affects only WAT browning but not interscapular BAT browning.
As shown in the Results (Figure 1), we compared BAT, visceral WAT (vWAT) and inguinal WAT weights between SPL KO and WT mice. The BAT mass and BAT weight/body weights were similar between SPL KO and WT mice (Figure 1 D&E). In contrast, SPL KO mice had significantly smaller vWAT and iWAT weights and fat/body weights, compared to corresponding WAT from WT controls (Fig 1D &E). We suspect that WAT browning contributes to reduced white adipose mass. Because there was a large reduction in vWAT weight and our previous study showed that the expression levels of browning markers were not significantly different in iWAT between SPL KO and WT mice (18), we focused on investigating the browning marker gene expression, mitochondria content number, and mitochondrial activity in vWAT from SPL KO and WT mice.
- The authors can do cold tolerance test for SPL KO mice if SPL plays a critical role in thermogenesis.
Thanks for the valuable suggestion, we add this to the discussion as a future plan.
- I suggest that the authors exclude PPAR-γ from their conclusions since they don’t have molecular mechanism between SPL and PPAR-γ. For example, the manuscript’s title can be “Deletion of Spinophilin Promotes White Adipocyte Browning”.
Changed as suggested.
Reviewer 2 Report
In this manuscript, Gou et al. the authors follow up on their previous work using whole-body KO mice of SPL. The authors confirm that constitutive whole-body deletion of SPL leads to the browning of visceral WAT and extent these observations to primary and 3T3-L1 cells, pointing to a cell-autonomous effect. This part of the paper is generally convincing. However, the mechanistic part and what we learn about SPL are very limited. Given the paper is a follow-up on an already published browning/thermogenesis phenotype, I was expecting a clearer mechanism of how SPL is repressing thermogenic genes and I think this part of the paper needs more investigation.
1) The authors present that SPL deletion induces a thermogenic program accompanied by more mitochondria etc. Experiments to elaborate on how SPL works as a repressor of the thermogenic gene program are needed.
2) The authors show that the deletion of SPL leads to more PPARy in the nucleus. Do the authors envision SPL retains PPARy in the cytoplasm? Direct interaction could be detected by co-IP studies.
3) Would overexpression of SPL block rosiglitazone-induced browning?
4) A model figure at the end of the paper summarizing a proposed mechanism would be beneficial.
5) Looks like the body length is lower in SPL KO mice in Figure 1C. The authors should quantify this, and if changed the body/tissue weight should be normalized.
Author Response
- The authors present that SPL deletion induces a thermogenic program accompanied by more mitochondria etc. Experiments to elaborate on how SPL works as a repressor of the thermogenic gene program are needed.
Great suggestion. However, how SPL function during thermogenesis will be investigated in future studies and reported separately.
- The authors show that the deletion of SPL leads to more PPARy in the nucleus. Do the authors envision SPL retains PPARy in the cytoplasm? Direct interaction could be detected by co-IP studies.
As shown in Figure 4 A&B, PPARg levels in both the nuclear and cytoplasm were higher in the KO cells compared to WT cells, although the difference were more dramatic in the nuclear than in the cytoplasma.
- Would overexpression of SPL block rosiglitazone-induced browning?
Great point, we can investigate this possibility and report the data separately.
- A model figure at the end of the paper summarizing a proposed mechanism would be beneficial.
A graphic abstract has been added as suggested.
5) Looks like the body length is lower in SPL KO mice in Figure 1C. The authors should quantify this, and if changed the body/tissue weight should be normalized.
That’s a thoughtful comment. We measured the body lengths of male SPL KO and wild type mice at 8 or 12 weeks of age, and there were no differences in body length observed (Fig 1).

Reviewer 3 Report
Title: Deletion of Spinophilin Promotes White Adipocyte Browning via Regulation of PPAR-γ
Authors: Wenyu Gou, Hua Wei, Lindsay Swaby, Erica Green, Hongjun Wang
General comment:
In the world struggling with the pandemic of obesity, there is a constant need for basic research that may provide potential therapeutic targets for disease treatment. In their work, Wenyu Gou et al. investigated the therapeutic potential of systemic spinophilin (SPL) deletion to reduce excess adiposity. They found that visceral white adipose tissue (vWAT) from SPL KO is characterized by a higher expression of brown adipocyte markers and higher mtDNA content compared to vWAT from wild-type (WT) animals. They also found that the molecular mechanism involved in the described phenomena includes regulating PPAR-γ activity. The study design is convincing, and the methodology is appropriately chosen; therefore, I have only some minor remarks the authors should consider before the manuscript is accepted for publication.
Minor revisions:
Results:
1) In Figures 1-3 legends, the Authors write, "Unpaired Student's t-test was performed to compare the difference between WT and KO. * p<0.05, **p<0.01.” With a relatively small number of studied animals, it is unlikely to obtain a normal data distribution. Please indicate if the data distribution was normal and if parametric tests were justified.
2) Figures 1-6 “All data are presented as means ± SEM” – if it occurs that the distribution of the analyzed parameters was not normal, the data should be preferably presented as medians with quartiles.
Discussion:
1) Please consider any other possible limitations that may reduce the therapeutic application of the study findings.
Author Response
Minor revisions:
Results:
- In Figures 1-3 legends, the Authors write, "Unpaired Student's t-test was performed to compare the difference between WT and KO. * p<0.05, **p<0.01.” With a relatively small number of studied animals, it is unlikely to obtain a normal data distribution. Please indicate if the data distribution was normal and if parametric tests were justified.
We appreciate the comment. We agree that a small number of animals may have a high chance of getting an unnormal distribution. We did pre-analysis for normal (Gaussian) distribution with QQ-plot and Shapiro-Wilk tests and found out that if we include >8 mice in a group, our data would fit a normal distribution. Therefore, most of our results were sampled from normally distributed data since we had enough replications or animals in each experiment. We understand that many statistical tests, such as ANOVA, linear regression, and t-test, assume data are sampled from a Gaussian distribution. Besides, we transformed the values to Logs to make the distribution more Gaussian for statistical significance analysis if our samples were not normally distributed. Also, we run an F-test to check if two populations have the same variance before the t-test.
- Figures 1-6 “All data are presented as means ± SEM” – if it occurs that the distribution of the analyzed parameters was not normal, the data should be preferably presented as medians with quartiles.
We appreciate the reviewer’s critical point on the statistical approach to the data analysis and presentation. The general guidance about whether to use median or mean was that if the data contains outliers then one should rather use the median because otherwise the value of the mean would be dominated by the outliers rather than the typical values. In this study, we plotted a histogram of the data, if we didn’t find outliers we used the mean.
Discussion:
1) Please consider any other possible limitations that may reduce the therapeutic application of the study findings.
Great suggestions, we have added additional limitations in the revised manuscript.
Reviewer 4 Report
The manuscript of “Deletion of Spinophilin Promotes White Adipocyte Browning via Regulation of PPAR-γ” by Wenyu Gou and co-authors aims to study the role of spinophilin (SPL) in the process of white adipocyte browning that may be a promising therapeutic approach to induce energy expenditure and counteract obesity and its complications. The authors showed that the depletion of SPL led to increased white adipocyte browning in in vitro and in vivo experiments. In SPL KO mice, the deletion of SPL led to induction of expressions of brown adipocyte-related genes, including UCP1, FABP4, and CIDEA in in visceral white adipose tissue (vWAT). In other series of experiments, the authors demonstrated that higher UCP1 expression in SPL KO cells was associated with the induction and nuclear tentation of PPAR-γ after rosiglitazone treatment, suggesting SPL could negatively control UCP1 expression via PPAR-γ, CIDEA, PRDM16. The authors suggested that targeting SPL can be used to promote WAT browning and reduce obesity.
The topic of the manuscript is highly relevant and timely in view of recent the statistics on the incidence of obesity and its associated complications in the world. The study was carried out at a high level using advanced approaches of molecular biology and is quite detailed. The manuscript is well structured; all the conclusions are supported by the data obtained. The manuscript makes a significant contribution to the systematization of knowledge about the molecular mechanisms of SPL action on the regulation of WAT browning.
Minor comments:
Lines 49-50: The term “protein phosphate-1 (PP1)” should be replaced by protein phosphatase 1 (PP1).
In the work, the authors generated SPL knockdown cells by transfecting 3T3L1 cells with the shRNA for SPL. However, the results of the RT-PCR and/or WB analysis to confirm the gene knockdown were not presented. The same applies to KO mice.
In the legends to Figs. 3-5 and in the Materials and methods section, it is necessary to specify the numbers of biological repeats in each group or set of experiments.
In the Discussion section, it would be good to discuss in a comparative aspect the contribution of each of the three proposed mechanisms of SPL action in vWAT to its overall effect.
Author Response
please check in attachment

Round 2
Reviewer 1 Report
I hope that your work will continue the drug targeting of SPL for metabolic diseases in future studies. It's considered suitable for the publication of the revised manuscript in "Pharmaceuticals".
Author Response
We thank you for the encouragement.
Reviewer 2 Report
- The authors present that SPL deletion induces a thermogenic program accompanied by more mitochondria etc. Experiments to elaborate on how SPL works as a repressor of the thermogenic gene program are needed.
Reply: Great suggestion. However, how SPL function during thermogenesis will be investigated in future studies and reported separately.
Reply reviewer: Given that the authors previously have published the same browning phenotype in the same mouse model (doi: 10.1152/ajpendo.00114.2020), I think the authors need to unfold the mechanistic part more.
- The authors show that the deletion of SPL leads to more PPARy in the nucleus. Do the authors envision SPL retains PPARy in the cytoplasm? Direct interaction could be detected by co-IP studies.
Reply: As shown in Figure 4 A&B, PPARg levels in both the nuclear and cytoplasm were higher in the KO cells compared to WT cells, although the difference were more dramatic in the nuclear than in the cytoplasma.
Reply reviewer: To me it is still unclear how the authors envision SPL to retain PPARy in the cytosol, when SPL is not knocked out (as shown in the graphical abstract for WT cells). What is the mechanism? Does SPL and PPARy interact in the cytosol? If yes, the authors have not shown it. If not, how does the presence of SPL then retain PPARy in the cytosol?
- Would overexpression of SPL block rosiglitazone-induced browning?
Reply: Great point, we can investigate this possibility and report the data separately.
Reply reviewer: I think the authors need to unfold the mechanistic part more. This line of experiments could be one way of doing it.
- A model figure at the end of the paper summarizing a proposed mechanism would be beneficial.
Reply: A graphic abstract has been added as suggested.
Reply reviewer: Great! However, I think the graphical abstract for WT cells is unclear. How does the authors envision SPL to retain PPARy in the nucleus. This is not obvious from the abstract or associated text.
5) Looks like the body length is lower in SPL KO mice in Figure 1C. The authors should quantify this, and if changed the body/tissue weight should be normalized.
Reply: That’s a thoughtful comment. We measured the body lengths of male SPL KO and wild type mice at 8 or 12 weeks of age, and there were no differences in body length observed (Fig 1).
Reply reviewer: Perfect!
Author Response
Attached please see point-to-point responses to the reviewer's comment. Thank you for making our manuscript better.

Round 3
Reviewer 2 Report
Sorry for being stubborn. As new mechanistic data is not added, the authors should at least for the graphical abstract and/or the discussion propose a mechanism of how the authors envision SPL to retain PPARy in the nucleus. This is not obvious in the current manuscript.
Author Response
We appreciated your valuable comments that made our manuscript better. We added the following discussion that can explain the effect of SPL. Please let us know if there is anything else you may need. Thank you.
Hongjun
Based on our data, the loss of SPL leads to an increased expression of PPARγ in both the cytosol and nuclei of vWAT adipocytes. This was the case in the quiescent stage as well as after treatment with Rosi, suggesting that SPL itself negatively regulates the expression level of PPARγ. The degradation of PPARγ after ligand activation is achieved via the ubiquitination-proteasome pathways [1]. Our data also showed that there was much more PPARγ remaining inside the nucleus of SPL KO cells at 48hr after Rosi treatment compared to WT cells, indicating that SPL is required for PPARγ degradation. Indeed, SPL was shown to directly regulate protein stability and/or ubiquitination in a cyclin-dependent kinase 5 (Cdk5)-dependent manner in Parkinson’s disease [2]. The ERK/Cdk5 axis also controls PPAR-γ function in type 2 diabetes [3]. Therefore, the loss of SPL preserved PPARγ protein stability and blocked its ubiquitination/degradation in the KO cells, likely in a Cdk5-dependent manner. However, further studies are needed to definitively reveal this mechanism.
References:
- Hauser, S.; Adelmant, G.; Sarraf, P.; Wright, H.M.; Mueller, E.; Spiegelman, B.M. Degradation of the peroxisome proliferator-activated receptor gamma is linked to ligand-dependent activation. J Biol Chem 2000, 275, 18527-18533.
- Edler, M.C.; Salek, A.B.; Watkins, D.S.; Kaur, H.; Morris, C.W.; Yamamoto, B.K.; Baucum, A.J., 2nd. Mechanisms regulating the association of protein phosphatase 1 with spinophilin and neurabin. ACS Chem Neurosci 2018, 9, 2701-2712.
- Banks, A.S.; McAllister, F.E.; Camporez, J.P.; Zushin, P.J.; Jurczak, M.J.; Laznik-Bogoslavski, D.; Shulman, G.I.; Gygi, S.P.; Spiegelman, B.M. An erk/cdk5 axis controls the diabetogenic actions of ppargamma. Nature 2015, 517, 391-395.
Round 4
Reviewer 2 Report
I think this is fine :)